# Anti-SARS-CoV-2 Agents in *Artemisia* Endophytic Fungi and Their Abundance in *Artemisia vulgaris* Tissue

**DOI:** 10.3390/jof9090905

**Published:** 2023-09-05

**Authors:** Shoji Maehara, Shogo Nakajima, Koichi Watashi, Andria Agusta, Misato Kikuchi, Toshiyuki Hata, Kento Takayama

**Affiliations:** 1Faculty of Pharmacy and Pharmaceutical Sciences, Fukuyama University, Sanzo,1 Gakuen-cho, Fukuyama, Hiroshima 729-0292, Japanhata@fukuyama-u.ac.jp (T.H.); takayama@fukuyama-u.ac.jp (K.T.); 2Department of Virology II, National Institute of Infectious Diseases, Toyama 1-23-1, Shinjuku-ku, Tokyo 162-8640, Japan; snakajima@niid.go.jp (S.N.); kwatashi@niid.go.jp (K.W.); 3Research Center for Drug and Vaccine Development, National Institute of Infectious Diseases, Toyama 1-23-1, Shinjuku-ku, Tokyo 162-8640, Japan; 4Research Center for Pharmaceutical Ingredients and Traditional Medicine, National Research and Innovation Agency (BRIN), Jalan Raya Bogor Km 46, Cibinong, Bogor 16911, Indonesia; andr005@brin.go.id; 5Medical Faculty, Malahayati University, Jl. Pramuka No.27, Kemiling Permai, Bandar Lampung 35152, Indonesia

**Keywords:** endophytic fungi, SARS-CoV-2, *Artemisia*, antivirus, endophytic abundance, metabarcoding

## Abstract

Coronavirus disease 2019 is caused by severe acute respiratory syndrome coronavirus 2 (SARS-CoV-2). Therapeutic agents for the disease are being developed. Endophytes are diverse and produce various secondary metabolites and bioactive substances. We isolated 13 endophytes from the leaves and stems of *Artemisia vulgaris*. Antiviral testing using the culture extracts of these endophytic fungi revealed that five isolates effectively inhibited the replication of SARS-CoV-2. These extracts were used to study the inhibitory effect of SARS-CoV-2 on 3C-like protease, and two isolates proved useful. Both isolates were from the genus *Colletotrichum*; therefore, the percentage of *Artemisia* endophytic fungi in the plant tissue was observed to be an important factor in plant site selection. Thus, we conducted a macroanalysis using next-generation sequencing to analyze the percentage of endophytes in the stems (whole, skin, and inner), leaves, roots, and cultivating soil, as well as to determine the location of each genus. To the best of our knowledge, this study is the first to report that *Colletotrichum* spp. are abundant in stems and that stem-based methods are the most efficient for isolating endophytes targeting *Colletotrichum* spp.

## 1. Introduction

Coronavirus disease 2019 (COVID-19), caused by severe acute respiratory syndrome coronavirus 2 (SARS-CoV-2), has been detrimental to global public health, the global economy, and society. Drugs that inhibit SARS-CoV-2 are clinically available, including neutralizing antibodies against viral spike proteins, protease inhibitors, and polymerase inhibitors. However, since variants can acquire resistance to some of the aforementioned drugs, new strategies are required to combat this virus.

Planet Earth has several symbioses from different life forms, including animals, plants, microorganisms, and humans. Plants live in symbiosis with other living organisms, providing them with food, fuel, oxygen, and other benefits. Endophytes are microorganisms that live in symbiosis with plants without harming their host plant [1,2]. Endophytes are beneficial to their hosts in terms of resistance to the environment [3] and reduced feeding damage [4,5]. In addition, endophytes contribute to the host and environmental organisms through the production of secondary metabolites and derivatization [6,7,8,9,10].

Throughout human history, microbial secondary metabolites, such as penicillin and streptomycin, have played important roles as specific drugs against infectious diseases. Microorganisms must survive and compete for survival. The risk of the outbreak of emerging infectious diseases, such as COVID-19, is increasing in this century because of climate change, urbanization, and global travel and trade. We have focused on the diversity of endophytes and their products as a method of combating these emerging infectious diseases.

The water extract of *Artemisia annua* has been shown to inhibit the replication of SARS-CoV-2 in vitro [11]. No reports have been published on the inhibition of SARS-CoV-2 replication using the extract of *Artemisia vulgaris* and its endophytic culture. However, it has been reported that endophyte products inhibit the replication of human coronavirus, HCoV229E [12]. The present study demonstrated the antiviral effect of endophytic products. 

*Artemisia* spp. are widespread in Asia, Europe, the Americas, and Africa, with a variety of more than 400 species. Thus, it is assumed that their endophytes are even more diverse, as has been reported [13]. However, the inhibitory effect of these *Artemisia*-endophyte-producing organisms on SARS-CoV-2 has not been reported. In 2017–2019, we identified the endophyte filamentous composition of *A. vulgaris* collected in Japan and Indonesia and reported the ability of the isolates to convert artemisinin [14]. 

Here, we prepared a culture extract of the endophyte filamentous fungi of *A. vulgaris* that inhibited the replication of SARS-CoV-2, measured the 3CL protease inhibitory activity of the extract, and discovered that genus *Colletotrichum* is a useful microbial strain. The next-generation sequencing (NGS) analysis of endophyte abundance in *A. vulgaris* tissue was employed to determine the proportion of *Colletotrichum* and other genera of filamentous fungi in *Artemisia* plants.

## 2. Materials and Methods

### 2.1. Host Plant and Its Endophytic Fungi

In 2019, a native *A. vulgaris* plant was collected at Fukuyama University (Hiroshima, Japan) after identification as a host plant, and the leaves and stems were obtained. The plant parts were used to isolate endophytes to obtain 13 filamentous species, including 5.8 S ribosomal DNA (rDNA). The sequence analysis of the internal transcribed spacer (ITS) 1 to ITS2 regions revealed the composition of the genus [14].

### 2.2. Preparation of Endophytic Culture Extracts 

Each of the 13 endophytic fungi isolated from *A. vulgaris* was inoculated in a potato dextrose broth (PDB) medium (200 mL; 24 g of PDB in 1 L of water) and incubated for 14 days (140 rpm reciprocating shaking, 27 °C). After homogenizing the culture medium, they were extracted with ethyl acetate (300 mL, three times). The extract was evaporated under reduced pressure and allowed to dry. Dimethyl sulfoxide (DMSO) was added to each extract to a concentration of 10 mg/mL and used as samples for the activity test against SARS-CoV-2.

### 2.3. Cell Culture

VeroE6/TMPRSS2 cells, VeroE6 cells overexpressing the transmembrane serine protease 2 gene, were cultured at 37 °C in 5% CO_2_, as previously described [15].

### 2.4. Infection Assay

SARS-CoV-2 was handled at a biosafety level 3 (BSL3). For the infection assay, VeroE6/TMPRSS2 cells were inoculated with a SARS-CoV-2 Wk-521 strain (2019-hCoV/Japan/TY/WK-521/2020) at a multiplicity of infection of 0.003 for 1 h and washed out. Cells cultured for 24 h were used to measure the extracellular viral RNA or those cultured for 48 h were examined for cytopathic effect (CPE). The 13 culture extracts from *Artemisia* endophytes were added during the virus inoculation (1 h) and post inoculation (24 or 48 h) at 100 and 33 µg/mL. Remdesivir (10 µM) was used as a positive control to inhibit virus replication.

### 2.5. Quantification of Viral RNA

Viral RNA was extracted using a MagMAX Viral/Pathogen II Nucleic Acid Isolation Kit (Thermo Fisher Scientific, Waltham, MA, USA) and quantified by real-time reverse-transcription polymerase chain reaction (RT-PCR) analysis with a one-step quantitative RT-PCR kit (THUNDERBIRD™ Probe One-step qRT-PCR kit, Toyobo, Osaka, Japan) as previously described [15].

### 2.6. Cell Viability

Cell viability was examined using Cell Counting Kit-8 (Dojindo Laboratories, Kuma-moto, Japan) as previously described [15].

### 2.7. The Ethanol Extract of A. vulgaris

The overground parts of *A. vulgaris* were collected at Fukuyama University. The plants were immediately freeze-dried and pulverized in a mixer. Five grams were weighed and extracted with 50 mL of ethanol for 24 h. The extract was concentrated in an evaporator for assay of bioactivity.

### 2.8. SARS-CoV-2 3C-like Protease Inhibitory Activity Test

The activity was determined using the Untagged 3C-like (3CL) Protease (SARS-CoV-2) Assay Kit (BPS Bioscience Inc., San Diego, CA, USA) according to the operating manual. The procedure is as follows. Here, 3CL protease (untagged enzyme, 4 ng/µL) in an assay buffer (30 µL) was preincubated with the test compounds (10 µL; final concentrations of 0.1, 0.3, and 1.0 mg/mL in a final volume of 50 µL) for 30 min at 25 °C. The enzymatic reaction was initiated by adding the substrate solution (10 µL). The reaction mixture was incubated for 4 h at 25 °C. Wells with 1% DMSO, 4 ng/µL enzyme, and 50 mM substrate functioned as positive controls with no enzyme inhibition. Wells with GC376 (final concentrations of 2.5, 7.5, 25, 75, and 250 µM) functioned as the standard inhibitor and negative control. The fluorescence intensity of each reacted fraction was measured using an Infinite 200 Pro microplate-reading fluorimeter (Tecan Trading AG, Zurich, Switzerland). Fluorescence measurements were performed at excitation and fluorescence wavelengths of 360 and 460 nm, respectively. Based on the obtained fluorescence intensity, dose–response curves were created using the statistical software, R, to determine the half-maximal inhibitory concentration (IC_50_).

### 2.9. Metabarcoding Analysis of Fungi Abundance in A. vulgaris and Their Cultivar Soil Sample Collection and DNA Extraction

*A. vulgaris* was collected from the same locations as the endophyte isolation. The following procedure was employed for the samples of leaves, stems (whole, outer skin, and inner), and roots: they were freeze-dried and crushed in a mixer. Soil samples were also collected from the *A. vulgaris* cultivated soil and preserved by freezing. Each sample (50 mg, plant; 25 mg, soil) was mechanically lysed using the Micro Smash MS-100 (Tomy Seiko Co., Ltd., Tokyo, Japan). Total DNA from the lysate was extracted using a phenol/chloroform/isoamyl alcohol mixture, precipitated, and washed with ethanol. Thereafter, the DNA pellets were resuspended in Tris-ethylenediaminetetraacetic acid (EDTA) buffer with RNase A (Sigma-Aldrich, St. Louis, MO, USA), purified using the High Pure PCR Template Preparation Kit (Roche Diagnostics, Mannheim, Germany), and suspended in an elution buffer (200 μL) according to the manufacturer’s instructions.

### 2.10. Analysis of the Mycobiome Composition 

The ITS region, as the primary fungal barcode marker, was amplified using the Tks Gflex DNA Polymerase (Takara, Shiga, Japan) and the primers ITS1F (5′-ACACTCTTTCCCTACACGACGCTCTTCCGATCTGGTCATTTAGAGGAAGTAA-3′) and ITS2R (5′-GTGACTGGAGTTCAGACGTGTGCTCTTCCGATCTGCTGCGTTCTTCATCGATGC-3′). The PCR products were purified using the AMPure XP purification kit (Beckman Coulter, Brea, CA, USA) and indexed using the primers 2ndF (5′-AATGATACGGCGACCACCGAGATCTACAC-Index2-ACACTCTTTCCCTACACGACGCTCTTCCGATCT-3′) and 2ndR (5′-CAAGCAGAAGACGGCATACGAGAT-Index1-GTGACTGGAGTTCAGACGTGTGCTCTTCCGATCT-3′).

The indexed library was purified using the AMPure XP purification kit and quantified using the Qubit 4 Fluorometer (Thermo Fisher Scientific, Tokyo, Japan). All the samples were added to the multiplex pool at equimolar concentrations and sequenced using the Illumina MiSeq platform (Illumina, San Diego, CA, USA). The amplified libraries were paired-end sequenced using the MiSeq Reagent Kit v3 (600 cycles) and PhiX control v3 (Illumina). The community analysis of the reads was performed using QIIME2 (ver. 2021.4) [16]. Primer sequences were removed using the Cutadapt plugin in QIIME2 [17]. The reads were denoised and clustered based on amplicon sequence variants (ASVs) at a single-nucleotide resolution using the DADA2 plugin in QIIME2 [18]. The derived ASVs were taxonomically classified using a naïve Bayes classifier trained on reference sequences based on operational taxonomic units clustered using a 99% identity threshold in the UNITE (v8.2) database https://unite.ut.ee/ (accessed on 19 May 2023) [19]. 

The ASVs (10,000 reads) were used for the alpha diversity estimation of the observed features (observed ASVs), Faith’s phylogenetic diversity (PD), and Shannon index. Beta diversity metrics were calculated using the ASVs from each sample based on the weighted and unweighted UniFrac distances. alpha- and beta diversity visualizations were conducted using ggplot2 and reshape2 in R (v4.1.2).

### 2.11. Statistical Analyses

Alpha diversity was analyzed using Tukey’s test. Beta diversity was visualized using principal coordinate analysis. Permutational multivariate analysis of variance was employed to detect statistical differences in the mycobial community structure of the groups. Statistical significance was set at *p* < 0.05.

## 3. Results

### 3.1. Anti-SARS-CoV-2 Activity of Artemisia Endophyte Extract

Each of the 13 endophytes previously identified was cultured in a PDB medium (27 °C, shaking culture) for 14 days. Each culture was extracted using ethyl acetate. The extracts were concentrated under reduced pressure and used as samples for an anti-SARS-CoV-2 assay. Table 1 lists the amounts of culture extract obtained from the 13 endophyte cultures.

#### 3.1.1. Effect of Endophytic Extracts on the Cytopathology of SARS-CoV-2-Infected Cells

VeroE6/TMPRSS2 cells were treated with the culture extracts at two concentrations (100 and 33 µg/mL) during the SARS-CoV-2 inoculation for 1 h and post inoculation for 48 h to observe the cytopathology induced by SARS-CoV-2 propagation in cells. Compounds that inhibit SARS-CoV-2 infection/replication augment cell viability. As shown in Figure 1A, four culture extracts augmented cell viability, showing more than 20% cell viability of the virus-free (virus-) control, including H-1 (*Mollisia* sp.), O-1, and two species of *Colletotrichum* sp., E-2, and L-3. However, Figure 1B shows no predominant cell viability at a culture extract concentration of 33 µg/mL.

#### 3.1.2. Viral RNA Level of Endophytic Extracts to SARS-CoV-2-Infected Cells

To examine the effect on the viral RNA levels, VeroE6/TMPRSS2 cells were treated with the extract at 100 or 33 µg/mL with a SARS-CoV-2 inoculum for 1 h and post inoculation for 24 h to recover the culture media to quantify the SARS-CoV-2 RNA. Five culture extracts reduced the viral RNA levels to less than 20% of the DMSO-treated control when used at a concentration of 100 µg/mL (Figure 2A). These included B-2 (*Cryptococcus* sp.), H-1 (*Mollisia* sp.), O-1 (*Aspergillus* sp.), and E-2 and L-3 (*Colletotrichum* sp.). In addition, B-2 exhibited an apparent reduction in viral RNA levels at 33 µg/mL (Figure 2B).

#### 3.1.3. Cytotoxicity to VeroE6/TMPRSS2 Cells

To examine the toxicity of the extract to cells independent of infection, endophyte culture extracts at 100 or 33 µg/mL were added to the VeroE6/TMPRSS2 cells without infection and incubated at 24 h to quantify cell viability. A-1 (*Aspergillus* sp.), B-2 (*Cryptococcus* sp.), L-1 (*Diaporthe* sp.), and N-1 (*Aspergillus* sp.) showed a reduction in viability to less than 80% of the DMSO-treated control (Figure 3), suggesting that the observed activity to reduce the viral RNA levels of B-2 was due to the cytotoxic effect. This was consistent with their inability to inhibit the SARS-CoV-2-induced CPE. Oppositely, H-1, O-1, E-2, and L-3 were suggested to exhibit anti-SARS-CoV-2 activities without cytotoxicity.

#### 3.1.4. SARS-CoV-2 3CL Protease Inhibitory Activity Test

B-2, E-2, H-1, L-3, O-1, and the ethanol extract of *A. vulgaris* were active in the antiviral test against SARS-CoV-2. They were tested during the 3CL protease inhibition activity test. The results showed that the enzyme activities of E-2 and L-3 (IC_50_: 95 and 244 µg/mL, respectively) significantly reduced, indicating 3CL protease inhibitory activity. Interestingly, both were observed to be *Colletotrichum* spp. (Figure 4). 

We have reported that the endophytic fungus, *Colletotrichum* sp., whose 3CL protease inhibitory activity was confirmed above (E-2 and L-3), is frequently isolated from *A. vulgaris*, which is native to Japan and Indonesia [14]. A comparison of endophyte compositions in different native habitats showed that *Colletotrichum* sp. tends to be frequently isolated as a common genus (Figure 5).

The tissues of *A. vulgaris* in which *Colletotrichum* is abundant will be important for strategies of future isolation experiments. Thus, a metabarcoding analysis of the ITS region was performed to determine the percentage of fungi abundance in each *A. vulgaris* tissue.

### 3.2. Metabarcoding Analysis of Fungi Abundance in A. vulgaris and Their Cultivar Soil

As previously indicated, the composition of endophytes is dependent on the environment of the host plant. Thus, it is assumed that endophytes infiltrate plants via soil infection. Here, metabarcoding analyses were performed in the fungal ITS region by sampling the leaves, stems, and roots of *A. vulgaris*, as well as cultivar soil. Sordariomycetes is a class of *Colletotrichum* spp. with the inhibitory activity of the 3CL protease of SARS-CoV-2. This metabarcoding analysis revealed that Sordariomycetes exist in each tissue (stem, leaf, and root) of *A. vulgaris* (Figure 6). In addition, these results showed that the mycobiome of the soil and plant exhibited different patterns (Figure 6). A similar case was observed for the underground part, the roots. To the best of our knowledge, this finding is the first evidence that roots, the boundary between plant and soil, constitute the mycobiome of plants.

Next, the alpha diversities of *A. vulgaris* leaves; stems (whole, skin, and inner); and roots were compared. Alpha diversity was calculated using three indices (observed features, Faith’s PD, and Shannon index) based on NGS-analyzed sequences. The results showed that the species diversity of the stem (inner) was significantly higher in all indices (Figure 7). This suggests less competition for survival in the stem. This may be because the inner part of the plant tissue is better protected than the root, leaf, and stem skin, which are in constant contact with the environment on the plant tissue’s surface. In addition, the roots have more diversity compared with those of the plant parts in contact with the environment. This allows for a new hypothesis that the soil in the rhizosphere protects the root endophytes.

The overall compositions of the mycobiome of *A. vulgaris* leaves, stems (whole, skin, and inner), and roots were compared using beta diversity indices for the weighted and unweighted UniFrac distance. The results showed that when Unweighted UniFrac was used as the index, there was an overlap between the whole stem and the skin and leaf samples, while the stem inner and root samples significantly changed diversity (Figure 8). When weighted UniFrac was used as the index, stem (whole)–leaf, stem (whole)–stem (skin), and stem (whole)–stem (inner) samples were observed to be close in distance, whereas the other sample combinations significantly changed diversity (Figure 8). In particular, the root samples substantially changed diversity compared with other samples. This finding provides evidence that rhizosphere soils strongly influence the composition of root endophytes.

Furthermore, in previous endophyte studies [14], we isolated the following from *A. vulgaris*: (a) *Aspergillus*, (b) *Cladosporium*, (c) *Colletotrichum*, (d) *Cryptococcus*, (e) *Diaporthe*, and (f) *Penicillium*. The percentage abundance in each tissue was calculated for each genus (Figure 9).

(a)*Aspergillus* spp. are more abundant on the stem and leaves and skin than on the inner part of the stem.(b)*Cladosporium* spp. are more abundant on the stem and leaves, with a tenth of the abundance of *Aspergillus*, and are present in the inner part of the stem and the skin to the same extent.(c)*Colletotrichum* spp. are more abundant on the stem and leaves and account for the largest abundance of the six genera. They are more abundant on the skin than on the inner part of the stem.(d)*Cryptococcus* spp. are abundant in the leaves and have the lowest abundance (approximately 0.05%).(e)*Diaporthe* spp. are abundant in the leaves and roots, with a particularly high percentage in the roots.(f)*Penicillium* spp. are abundant in the stem and leaves, with a higher percentage abundance in the skin than in the inner part of the stem.

**Figure 9 jof-09-00905-f009:**
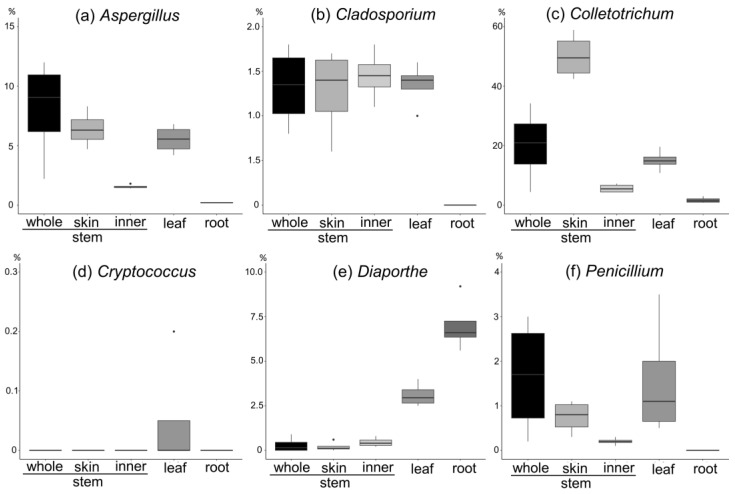
Fungi abundance of (**a**) *Aspergillus*, (**b**) *Cladosporium*, (**c**) *Colletotrichum*, (**d**) *Cryptococcus*, (**e**) *Diaporthe*, and (**f**) *Penicillium* in each *A. vulgaris* tissue.

## 4. Discussion

Here, we observed that the culture extracts of five symbiotic microorganisms (*Cryptococcus* sp. of B-2, *Mollisia* sp. of H-1, *Aspergillus* sp. of O-1, and *Colletotrichum* spp. of E-2 and L-3) isolated from *A. vulgaris* exhibited anti-SARS-CoV-2 activity in a cell culture infection model. As far as we know, this is the first report of anti-SARS-CoV-2 activity in culture extracts of *A. vulgaris* endophytes. Two extracts (*Colletotrichum* spp. of E-2 and L-3) exhibited 3CL protease inhibitory activity against SARS-CoV-2. Interestingly, two filamentous fungi of *Colletotrichum* were shown to target the 3CL protease of SARS-CoV-2. Filamentous fungi of the genus *Colletotrichum* are known plant pathogens that infect crops and cause anthracnose. Many *Colletotrichum* spp. are also producers of a variety of secondary metabolites with diverse biological activities [20]. The genus *Colletotrichum*, with its diverse production potential, is useful in the search for biologically active substances.

Metabarcoding analysis using NGS revealed significant differences between the mycobiome abundance in the stems, leaves, and roots of *A. vulgaris* and the mycobiome pattern in the cultivar soil. To date, the comparison of endophytes in plant tissues with the mycobiome of their culture soils has been the first challenge to visually demonstrate the differences. The highest ratio of Dothideomycetes was present in each tissue of *A. vulgaris*, whereas, in the soil, Agaricomycetes was the most present, forming a mycobiome. This was similar to our previous 2023 report on the endophyte composition of *A. vulgaris* [14] and the bar graph reconstructed with the endophyte composition class of *Artemisia* spp. published in 2018 [13]. More than 70% of the endophytes were composed of Dothideomycetes and Sordariomycetes, with less than 2% of Agaricomycetes (Appendix A). Thus, the visualization of the mycobiome derived from the present metabarcoding analysis was highly reproducible.

In addition, the alpha diversity analysis showed that the inner part of the plant was more advantageous for diversity in comparison with the plant part in contact with the outside and the interior. This was supported by Paloma et al. [21] who reported that the endosphere of *Arabidopsis thaliana* roots was more diverse than the episphere, attributable to the bacterial flora of the soil rhizosphere. The diversity of the fungal flora in the stem also increases the potential to benefit from the acquisition of fungus-derived secondary metabolites and their derivatives.

This finding can be rephrased as the reliability of the data on genus abundance in each tissue shown in Figure 9. Thus, if the aim is to obtain endophytes of genera *Diaporthe* and *Cryptococcus*, the survey should focus on roots and leaves. If the aim is to acquire *Aspergillus*, *Cladosporium*, *Colletotrichum*, and *Penicillium* genera, the stem and leaves should be mainly investigated. However, the genera, *Aspergillus*, *Penicillium*, and *Colletotrichum*, which had antiviral effects on SARS-CoV-2, are more abundant on the skin than in the stem. This suggests that when isolating the endophytes, surface sterilization should be carefully performed by, for example, lowering the concentration of sodium hypochlorite and shortening the soaking time. The study findings can serve as a guide for endophyte surveys in the future.

## 5. Conclusions

This research has indicated that the stem skins of *Artemisia vulgaris* are particularly effective in obtaining endophytes that can combat SARS-CoV-2. Specifically, *Colletotrichum* spp. found within the stem skins has been found to possess the ability to inhibit the 3CL protease of SARS-CoV-2, while *Aspergillus* sp. is effective against infected cells. One of the key benefits of this innovative method is that the target can be chosen by adjusting the specific part of the plant used for isolation. Leaves with a high abundance of *Penicillium* spp. are optimal for antibiotics, while roots with a high abundance of *Diaporthe* spp. [22] are optimal for antifungal and anti-inflammatory targets. This innovative study exemplifies a highly efficient system for discovering a diverse range of endophytes that can be evaluated for their potential in developing new drugs against various organisms.

## Figures and Tables

**Figure 1 jof-09-00905-f001:**
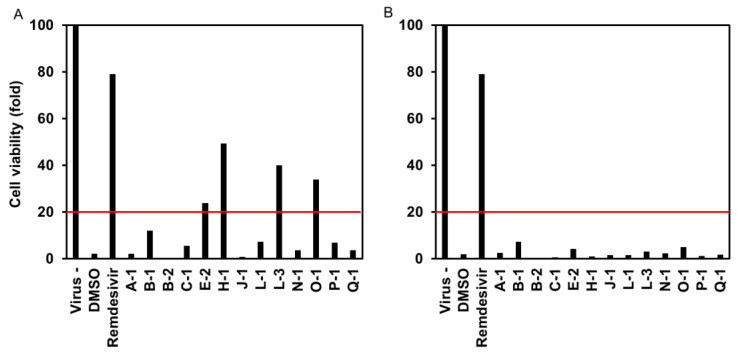
Protection by each endophytic extract from the SARS-CoV-2-induced cytopathic effect. (**A**,**B**) show cell viability with extract concentrations at 100 and 33 µg/mL, respectively. The red line indicates 20% cell viability.

**Figure 2 jof-09-00905-f002:**
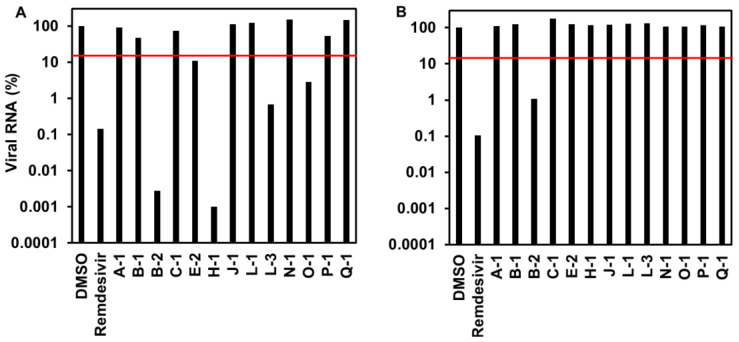
Viral RNA levels in the media of SARS-CoV-2-infected cells upon treatment with endophytic extracts at 100 (**A**) and 33 µg/mL (**B**). The red line indicates 20% viral RNA level.

**Figure 3 jof-09-00905-f003:**
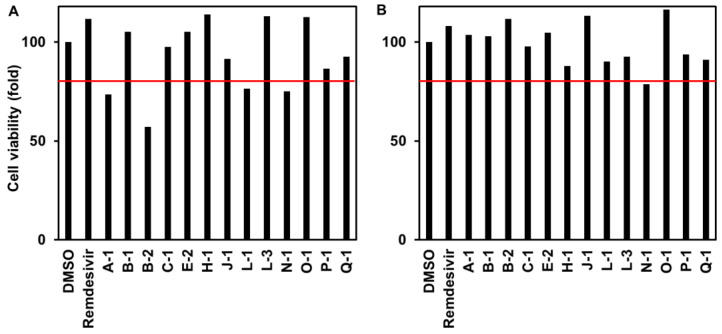
Cytotoxicity of each endophytic extract to VeroE6/TMPRSS2 cells (*n* = 1). The red line indicates 80% cell viability. This subfigure verifies that Figure 1 and Figure 2 show antiviral effects.

**Figure 4 jof-09-00905-f004:**
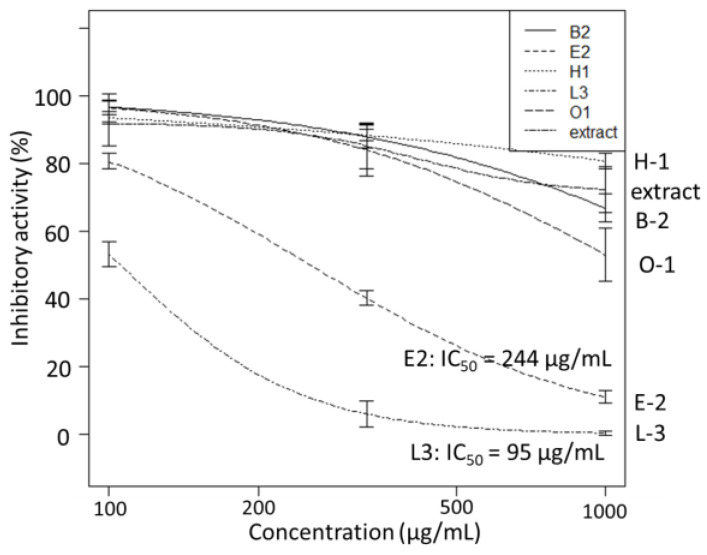
The 3CL protease inhibitory activity (*n* = 3) of endophyte extracts.

**Figure 5 jof-09-00905-f005:**
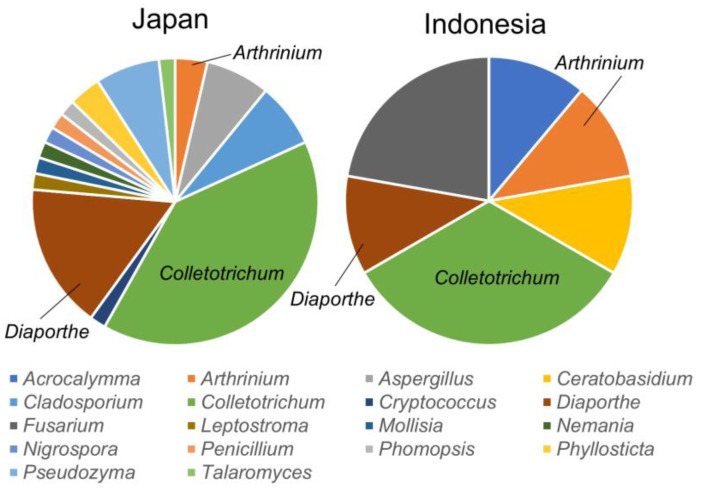
Comparison of isolated endophyte genus composition from *A. vulgaris* native to Japan (2017–2019) and Indonesia (2017).

**Figure 6 jof-09-00905-f006:**
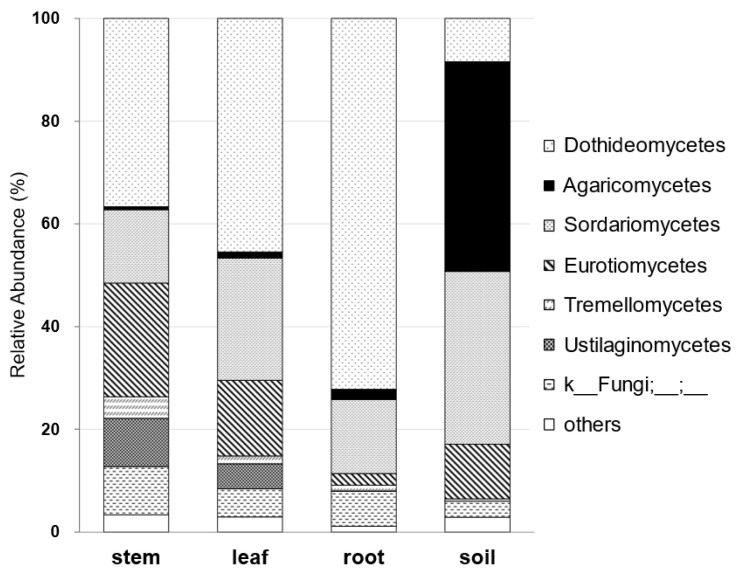
Comparison of class-relative fungus abundance (%) in the stem, leaf, and root of *A. vulgaris* and cultivar soil analyzed by metabarcoding.

**Figure 7 jof-09-00905-f007:**
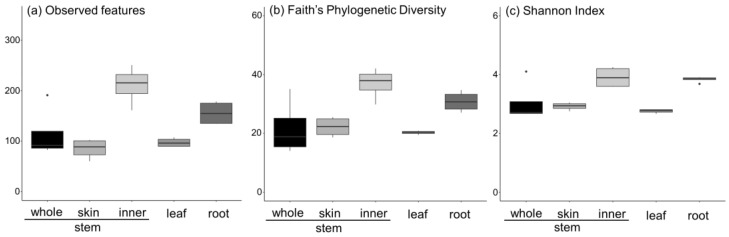
Alpha diversity comparison of the fungus abundance in *A. vulgaris* tissue was calculated using (**a**) observed features, (**b**) Faith’s phylogenetic diversity, and (**c**) Shannon index.

**Figure 8 jof-09-00905-f008:**
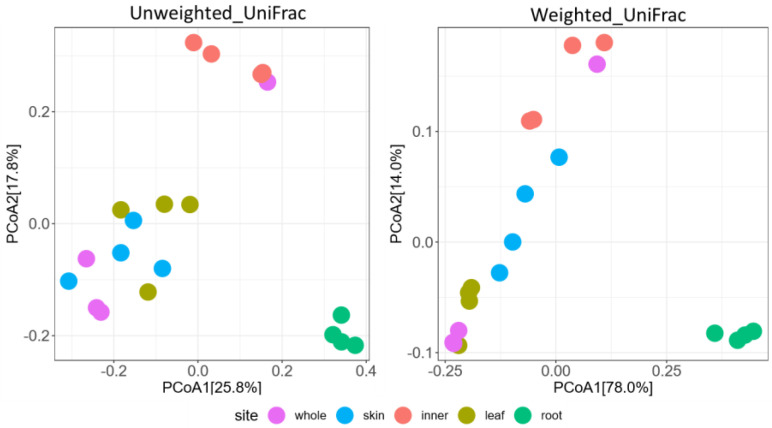
Community structure of fungus abundance in the stem (whole, skin, and inner), leaf, and root of *A. vulgaris* in 20 samples determined by principal coordinate analysis (PCoA). Two-dimensional PCoA results are plotted based on unweighted and weighted UniFrac distances.

**Table 1 jof-09-00905-t001:** Isolated endophytic fungi and extract weights with ethyl acetate.

Collection Name	Accession Number	Genus	Extract Weight (mg)
yomogi fungi A-1	LC719230	*Aspergillus* sp.	80
yomogi fungi B-1	LC719228	*Cladosporium* sp.	80
yomogi fungi B-2	LC719244	*Cryptococcus* sp.	40
yomogi fungi C-1	LC719242	*Penicillium* sp.	70
yomogi fungi E-2	LC719209	*Colletotrichum* sp.	50
yomogi fungi H-1	LC719243	*Mollisia* sp.	40
yomogi fungi J-1	LC719231	*Aspergillus* sp.	130
yomogi fungi L-1	LC719219	*Diaporthe* sp.	150
yomogi fungi L-3	LC719210	*Colletotrichum* sp.	80
yomogi fungi N-1	LC719232	*Aspergillus* sp.	70
yomogi fungi O-1	LC719233	*Aspergillus* sp.	40
yomogi fungi P-1	LC775239	*Hypomontagnella* sp.	120
yomogi fungi Q-1	LC719229	*Cladosporium* sp.	40

## Data Availability

Not applicable.

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
