# Peer review of "Anti-SARS-CoV-2 Agents in Artemisia Endophytic Fungi and Their Abundance in Artemisia vulgaris Tissue"

_jof, 2023, doi:10.3390/jof9090905_

Round 1
Reviewer 1 Report
Section 2.2 - were all 13 endophytic fungi grown separately on PDA? Please clarify. It reads as if that is the case but make it crystal clear.
Need to marry the findings together. Need to say where in the plant it is best to get endophytes that have activity against SARS-CoV-2. This really an important point as we need new drugs for SARS-COV-2.
The other point to discuss may be that using certain parts of the plant we may get other endophytes that have activity against other organisms.
This study provides for the first time an efficient system for finding different endophytes to examine for potential new drugs against different organisms.
Need to draw out these points/make them explicit.
Author Response
Reviewer #1:
Section 2.2 - were all 13 endophytic fungi grown separately on PDA? Please clarify. It reads as if that is the case but make it crystal clear.
>>The 13 isolates were cultivated separately. So, we revised manuscript L 83 to “Each of 13 endophytic fungi….”.
Need to marry the findings together. Need to say where in the plant it is best to get endophytes that have activity against SARS-CoV-2. This really an important point as we need new drugs for SARS-COV-2.
The other point to discuss may be that using certain parts of the plant we may get other endophytes that have activity against other organisms.
This study provides for the first time an efficient system for finding different endophytes to examine for potential new drugs against different organisms.
>>Thank you for the good advice. We added a conclusion to section 5 and mentioned that.
Need to draw out these points/make them explicit.
Reviewer 2 Report
Sentence L 69-71 is not clear. Further thin layer chromatography data are not given in the materials and methods and in the text. Usually, this method is used for preliminary analysis of secondary metabolites synthesized by strains, and not to show that microbial strains are not the same. It is necessary to either remove this sentence or describe it.
The legend for Figure 6 should be made larger, the designations are incomprehensible.
L216 “the ethanol extract of A. vulgaris” - indicate in the materials and methods how you received it.
L 275 – “in previous endophyte studies” - provide reference.
In the discussion, the first paragraph needs to be edited (L297-L301):
- “five symbiotic microorganisms” - list it.
Swap the sentences according to the meaning in the following order: 1, 3, 2, 4 (L297-L301). Next, add one or two sentences about known secondary metabolites of Colletotrichum and references, for example - doi: 10.1007/s12272-019-01142-z.
L. 235 - A. Vulgaris
When describing the metabarcoding analysis of fungi in A. vulgaris I recommend giving the name of the class of Colletotrichum and discussing its occurrence.
Author Response
Reviewer #2:
Sentence L 69-71 is not clear. Further thin layer chromatography data are not given in the materials and methods and in the text. Usually, this method is used for preliminary analysis of secondary metabolites synthesized by strains, and not to show that microbial strains are not the same. It is necessary to either remove this sentence or describe it.
>> We have deleted the relevant sentence (L 69-70) by the reviewer's suggestion.
The legend for Figure 6 should be made larger, the designations are incomprehensible.
>> We have enlarged the relevant parts of the figure by the reviewer's suggestion.
L216 “the ethanol extract of A. vulgaris” - indicate in the materials and methods how you received it.
>>We have added an experiment section (2.7. The ethanol extract of A. vulgaris).
L 275 – “in previous endophyte studies” - provide reference.
>>We have added a reference at L 275.
In the discussion, the first paragraph needs to be edited (L297-L301):
- “five symbiotic microorganisms” - list it.
>>We have listed five symbiotic microorganisms at L 314-315.
Swap the sentences according to the meaning in the following order: 1, 3, 2, 4 (L297-L301). Next, add one or two sentences about known secondary metabolites of Colletotrichum and references, for example - doi: 10.1007/s12272-019-01142-z.
>>We changed the sequence of sentences. Following that, we added a sentence about Colletotrichum and references.
- 235 - A. Vulgaris
>>We have revised it to A. vulgaris.
When describing the metabarcoding analysis of fungi in A. vulgaris I recommend giving the name of the class of Colletotrichum and discussing its occurrence.cript.
>>We have added sentences at L 250-253.